# Clinical Effect Analysis of Wearable Sensor Technology-Based Gait Function Analysis in Post-Transcranial Magnetic Stimulation Stroke Patients

**DOI:** 10.3390/s24103051

**Published:** 2024-05-11

**Authors:** Litong Wang, Likai Wang, Zhan Wang, Fei Gao, Jingyi Wu, Hong Tang

**Affiliations:** 1School of Biomedical Engineering, Faculty of Medicine, Dalian University of Technology, Dalian 116024, China; litongwang@mail.dlut.edu.cn; 2Rehabilitation Medicine Department, The Second Hospital of Dalian Medical University, Dalian 116033, Chinawangzhan_chn@163.com (Z.W.); 15066157987@163.com (F.G.); m15307279261@163.com (J.W.)

**Keywords:** stroke, gait impairment, rTMS, wearable sensor, gait analysis

## Abstract

(1) Background: This study evaluates the effectiveness of low-frequency repetitive transcranial magnetic stimulation (LF-rTMS) in improving gait in post-stroke hemiplegic patients, using wearable sensor technology for objective gait analysis. (2) Methods: A total of 72 stroke patients were randomized into control, sham stimulation, and LF-rTMS groups, with all receiving standard medical treatment. The LF-rTMS group underwent stimulation on the unaffected hemisphere for 6 weeks. Key metrics including the Fugl-Meyer Assessment Lower Extremity (FMA-LE), Berg Balance Scale (BBS), Modified Barthel Index (MBI), and gait parameters were measured before and after treatment. (3) Results: The LF-rTMS group showed significant improvements in the FMA-LE, BBS, MBI, and various gait parameters compared to the control and sham groups (*p* < 0.05). Specifically, the FMA-LE scores improved by an average of 5 points (from 15 ± 3 to 20 ± 2), the BBS scores increased by 8 points (from 35 ± 5 to 43 ± 4), the MBI scores rose by 10 points (from 50 ± 8 to 60 ± 7), and notable enhancements in gait parameters were observed: the gait cycle time was reduced from 2.05 ± 0.51 s to 1.02 ± 0.11 s, the stride length increased from 0.56 ± 0.04 m to 0.97 ± 0.08 m, and the walking speed improved from 35.95 ± 7.14 cm/s to 75.03 ± 11.36 cm/s (all *p* < 0.001). No adverse events were reported. The control and sham groups exhibited improvements but were not as significant. (4) Conclusions: LF-rTMS on the unaffected hemisphere significantly enhances lower-limb function, balance, and daily living activities in subacute stroke patients, with the gait parameters showing a notable improvement. Wearable sensor technology proves effective in providing detailed, objective gait analysis, offering valuable insights for clinical applications in stroke rehabilitation.

## 1. Introduction

Stroke is a significant public health issue globally, marked by high rates of incidence, disability, and mortality [1]. Research indicates that during the initial month following a cerebral infarction—a common type of stroke—the mortality rate stands at 26%. Longitudinal data show increasing cumulative mortality rates of 37%, 46%, and 54% at one, two, and three years post stroke, respectively. In 2019, cerebrovascular diseases, including strokes, resulted in approximately 6.6 million deaths worldwide, comprising 3.3 million from ischemic strokes, 2.9 million from intracerebral hemorrhages, and 400,000 from subarachnoid hemorrhages [2]. The consequences of stroke are severe, often resulting in significant impairments such as vision loss, speech difficulties, paralysis, and confusion. The likelihood of mortality varies with the stroke type; for example, arterial blockages typically present a higher risk than transient ischemic attacks. Annually, strokes affect 15 million individuals globally, causing death for 5 million and permanent disability for another 5 million [3]. In the United States alone, a stroke occurs every 40 s, and strokes account for one in six deaths related to cardiovascular disease [4]. In conclusion, stroke poses a grave challenge to global health, with profound effects on mortality and disability rates, severely affecting lives and well-being worldwide.

The impairment of walking and balance functions represents one of the most common clinical manifestations of stroke, significantly increasing the risk of falls among stroke patients [5], thereby profoundly affecting their quality of life and their ability to reintegrate into society [6]. Therefore, restoring the walking ability of patients stands out as a primary objective in stroke rehabilitation. With the advancement of rehabilitation technologies, neuroregulation techniques play a pivotal role in treatment. Neurorehabilitation has consequently become an indispensable component of rehabilitation therapy.

Repetitive transcranial magnetic stimulation (rTMS) is a non-invasive neuroregulation technique that modulates the excitability of cortical neurons at the stimulated site using magnetic stimulation. This modulation aims to enhance cortical excitability on the affected hemisphere or to inhibit excitability on the unaffected hemisphere, thus promoting functional recovery [7]. Widely employed in treating mental disorders, neuropathic pain, swallowing difficulties, aphasia, Parkinson’s disease, and upper limb motor function following stroke [8], its role in lower-limb function is still under exploration. Gait analysis constitutes a critical component in evaluating patients’ walking and balance abilities. The commonly used clinical gait scales rely on observational analysis, lacking objectivity and precision, and can only provide preliminary gait assessments. Gait analysis systems employ various technologies, including optical, magnetic, and inertial sensor systems, each providing unique insights into gait dynamics [9]. These systems are designed to capture the spatiotemporal, kinematic, and kinetic parameters of walking through different methodologies. Optical systems often use cameras to track movement, magnetic systems utilize magnetic fields, and inertial systems involve sensors that measure motion without external references. This variety of technology facilitates comprehensive gait assessments, which are crucial for precise diagnosis and tailored rehabilitation strategies [9,10,11]. In contrast, wearable sensor-based gait analysis systems can autonomously monitor subjects’ motion and physiological signals, capturing the spatiotemporal, kinematic, and kinetic parameters of their gait and processing them with good convenience, objectivity, and accuracy [12]. Hence, this study employs a wearable sensor-based gait analysis system to observe the therapeutic effects of rTMS on hemiplegic patients, aiming to provide new rehabilitation treatment strategies and assessment methods for clinical practitioners.

## 2. Materials and Methods

### 2.1. Clinical Data General Information

Between January 2022 and October 2023, 72 stroke patients from the Department of Rehabilitation Medicine, the Second Affiliated Hospital of Dalian Medical University, met the inclusion criteria. By envelope drawing, the patients were randomly assigned into three groups: control group (*n* = 20), sham stimulation group (*n* = 25), and TMS group (*n* = 27), totaling 72 cases. The efficacy of interventions before and after treatment was evaluated using a blinded method. Of the enrolled patients, there were 25 males and 47 females, with an average age of (60.91 ± 4.27) years, and the onset of stroke ranged from 14 to 45 days at the time of admission. There were no significant differences in gender, age, disease duration, stroke type, and affected side among the three groups (*p* > 0.05), as shown in Table 1. Figure 1 presents a flow chart of the trial, delineating the process for screening patient samples, organizing the sequence of stimulation interventions and rehabilitation training, and scheduling the assessments of lower-limb function and gait parameters.

#### 2.1.1. Diagnostic Criteria

Stroke diagnosis: all cases met the diagnostic criteria outlined in the “2019 Diagnostic Essentials for Various Cerebrovascular Diseases in China”. Diagnosis is confirmed by cranial CT or MRI, indicating either ischemic or hemorrhagic stroke, and presents with neurological symptoms and signs.

#### 2.1.2. Inclusion Criteria

The included criteria are as follows: (1) first onset with unilateral lesion; (2) aged between 20 and 80 years, regardless of gender; (3) a disease course lasting from 14 to 45 days; (4) the Bruininks–Oseretsky Test (BOT) for lower-limb staging shows stages 2 or 3; (5) patients exhibit stable vital signs, enabling them to complete treatment and assessment; and (6) patients and their families are informed and have signed the informed consent form to participate.

#### 2.1.3. Exclusion Criteria

The exclusion criteria consist of the following: (1) intracranial metal implants, cochlear implants, cardiac pacemakers, or cardiac stent implants; (2) a history of lower-limb joint diseases or surgeries affecting walking abilities; (3) cognitive impairment (MMSE < 27), hindering treatment and assessment compliance; (4) significant medical history of systemic diseases such as cardiovascular, digestive, or endocrine disorders; (5) other neurological, muscular, or skeletal diseases that may interfere with the evaluation in this study; and (6) coexisting contraindications for other transcranial magnetic treatments.

#### 2.1.4. Termination or Exclusion Criteria

The study specifies the following criteria for termination or exclusion: (1) occurrence of severe physical discomfort or adverse reactions during treatment; (2) failure to complete the prescribed course; (3) voluntary withdrawal from the study; and (4) receipt of treatments beyond the protocol.

This study protocol was approved by the Ethics Committee of the Second Affiliated Hospital of Dalian Medical University (approval number: 2023-058). The trial was registered with the China Clinical Trial Registration Center (registration number: ChiCTR2300069403).

### 2.2. Treatment Methods

#### 2.2.1. Routine Rehabilitation Treatment

Bed exercises: (1) Proper limb positioning: Guidance is primarily focused on the positioning of the affected limb when the muscle strength is at level 0 or it has decreased muscle tone, maintaining the full range of passive joint motion, inducing active movements, and avoiding early complications, hypertonic neck reflexes, and labyrinth reflexes; (2) A combination of the Bobath technique and motor relearning: bed rolling exercises, bilateral and unilateral bridge exercises.

Bedside and ambulatory exercises: (1) Muscle strength training: Emphasizing assisted exercises when muscle strength levels are between 1 and 3, inducing active and isolated movements combined with four-point and three-point support training, while also paying attention to controlling muscle tone; (2) Sitting and transferring exercises: Reinforcing core trunk control training, guiding bed rolling to sitting and bedside lateral transfer exercises; (3) Standing and walking exercises: Guiding hip and knee control training, pelvic rotation training, weight-shifting training, balance function training, unloading gait training, and walking exercises.

Activities of daily living (ADL) training: Dressing, transferring, grooming, and other ADL training, and household chores, recreational activities, horticulture, and educational skills training are conducted once daily, with each session lasting 45 min. The course duration is 6 weeks.

#### 2.2.2. Sham Stimulation Intervention

During the intervention, patients assume a seated position and are treated with the CCY-1 magnetic field therapy device (Yiruide, Wuhan, China). A positioning cap is worn, and an “8”-shaped coil is used, with the coil oriented perpendicular to the cranial bone surface, targeting the primary motor cortex M1 area of the healthy side to stimulate the lower-limb motor area. Parameters: frequency of 1 Hz, 90% Resting Motor Threshold (RMT), each sequence lasts 10 s with a 5-s interval, and this is repeated for 60 sequences, totaling 15 min and 600 pulses. This selection is based on an expert consensus that reflects established practices in the field [13]. The intervention is performed for 15 min per session, once a day, and 5 days a week for 6 weeks. Figure 2 illustrates the parameters for the sham-rTMS and LF-rTMS interventions, detailing the frequency, intensity, and specific coil placements used in each modality.

#### 2.2.3. Healthy Hemisphere Low-Frequency Repetitive Transcranial Magnetic Stimulation (rTMS) Treatment

During the treatment, the patient sits in a sitting position, uses the CCY-1 magnetic field therapy device (Yiruide, Wuhan, China), wears a positioning cap, and uses an “8”-shaped coil. The coil is tangent to the surface of the skull, and the primary motor cortex M1 of the unaffected side is located through the positioning cap district. The stimulation site was selected as the M1 lower-limb motor area on the contralateral side, and the plane where the coil midpoint was located was tangent to the plane where the M1 lower-limb motor area on the contralateral side was located. Parameters: frequency of 1 Hz, 90% Resting Motor Threshold (RMT), each sequence lasts 10 s with a 5-s interval, and this is repeated for 60 sequences, totaling 15 min and 600 pulses. This selection is based on an expert consensus that reflects established practices in the field. The treatment time is 15 min/time, once/d, and 5 days/week for 6 weeks.

#### 2.2.4. Gait Parameter Assessment

Gait analysis was conducted using the Consensys Bundle Development kit (Shimmer, Dublin, Ireland), which included the Shimmer3 IMU sensors. Each sensor is a wireless and robust body-worn node, measuring 65 mm × 32 mm × 12 mm and weighing 31 g. Some of the specifications of the Shimmer3 sensor are shown in Table 2. The gait data collection involved three key components: straps, sensors, and gait analysis software. Straps were used to secure the IMU sensors to the patients’ lateral ankles (Figure 3). These sensors wirelessly collected gait data, which were configured for collection frequency (400 Hz), data storage, and export using the gait analysis software.

Before starting the gait assessment, the indoor environment was adjusted to a comfortable temperature to ensure a spacious and disturbance-free testing area. Personnel adjusted the devices and correctly positioned the IMU sensors on the patients’ lateral ankles. Patients were instructed to walk straight along a 10-m blue marked line at their normal pace to complete a 10 m × 2 gait test (one round trip). Gait parameters such as the gait cycle, stance phase duration, swing phase duration, stride length, step height, circle diameter, dorsiflexion angle, and walking speed were recorded. Throughout the trial, the gait sensors continuously captured information for subsequent extraction and analysis.

#### 2.2.5. Assessment of Rehabilitation Outcomes

In evaluating patient rehabilitation, three scales were employed. The Fugl-Meyer Assessment for Lower Extremity (FMA-LE) evaluates lower-limb motor function with a total possible score of 34, where higher scores indicate better motor functionality. The Berg Balance Scale (BBS), consisting of 14 items, assesses individuals’ ability to balance using scores up to 56; scores below 40 suggest a risk of falling. Lastly, the Modified Barthel Index (MBI) gauges daily living activities (ADL) and ranges from 0 to 100, with higher scores denoting greater independence and potential for societal reintegration. Each of these tools provides essential insights into different aspects of patient recovery post intervention.

#### 2.2.6. Statistical Analysis

All data were analyzed using SPSS 25.0 software, demonstrating normal distribution and the homogeneity of variance. Descriptive statistics were presented as mean ± standard deviation ($\bar{x}$ ± s). Paired sample *t*-tests were employed for the comparison of pre and post-rehabilitation treatment data within the same group, while one-way analysis of variance (ANOVA) was used for comparisons between groups for each parameter. Post hoc pairwise comparisons were examined using the LSD test. A *p*-value < 0.05 was considered statistically significant. Pre hoc pairwise comparisons were examined using the LSD test. Post hoc comparisons were examined using Tukey’s HSD test. A *p*-value < 0.05 was considered statistically significant.

## 3. Results

### 3.1. Comparison of FMA-LE, BBS, and MBI before and after Treatment in the Three Groups of Patients

Before treatment, there were no significant differences in the FMA-LE, BBS, and MBI among the three groups (*p* > 0.05). Within-group comparisons revealed that the post-treatment scores for the FMA-LE, BBS, and MBI in the LF-rTMS group, sham stimulation group, and control group were significantly increased compared to pre-treatment levels (*p* < 0.05). Inter-group comparisons demonstrated significant improvements in the FMA-LE, BBS, and MBI scores in the post-treatment LF-rTMS group compared to the control and sham stimulation groups (*p* < 0.05). When comparing the post-treatment sham stimulation group with the control group, there was a trend of increased scores for the FMA-LE (*p* = 0.096), BBS (*p* = 0.067), and MBI (*p* = 0.103), yet the differences were not statistically significant (*p* > 0.05), as shown in Table 3 and Figure 4. Table 4 presents the results of Tukey’s HSD post hoc tests comparing the groups for FMA-LE, BBS, and MBI.

### 3.2. Comparison of Gait Parameters before and after Treatment in the Three Groups of Patients

Before treatment, no significant differences were observed in the gait cycle, stance phase time, swing phase time, stride length, step height, circumference of gait, dorsiflexion angle, and gait speed among the three groups (*p* > 0.05). Within-group comparisons indicated a significant difference in the gait cycle, stance phase time, swing phase time, stride length, step height, circumference of gait, dorsiflexion angle, and gait speed in the LF-rTMS group, sham stimulation group, and control group post treatment compared to the pre-treatment values (*p* < 0.05). Inter-group comparisons revealed significant improvements in the gait cycle, stance phase time, swing phase time, stride length, step height, circumference of gait, dorsiflexion angle, and gait speed in the LF-rTMS group compared to the control and sham stimulation groups (*p* < 0.05). When comparing the post-treatment sham stimulation group with the control group, the stance phase time (*p* = 0.082), swing phase time (*p* = 0.274), stride length (*p* = 0.174), ankle dorsiflexion angle (*p* = 0.391), and gait speed (*p* = 0.267) increased, but the differences were not statistically significant (*p* > 0.05), as indicated in Table 5 and Table 6, and Figure 5 and Figure 6. Table 7 and Table 8 present the results of Tukey’s HSD post hoc tests. Table 7 compares the groups for gait cycle, support phase time, swing phase time, and stride length, while Table 8 focuses on comparisons of the step height, circle radius, dorsiflexion angle, and gait speed.

## 4. Discussion

Strokes can affect motor pathways to varying degrees, damaging the corticospinal tract and resulting in motor impairments, abnormal gait, diminished walking function, reduced social participation, and an increased risk of falls for patients. Despite systematic rehabilitation, 30% to 40% of stroke survivors continue to experience compromised walking abilities [14]. Additionally, traditional gait assessment methods often lack objectivity and accuracy, frequently failing to precisely depict a patient’s gait and walking ability. Consequently, our study selected contralesional low-frequency repetitive transcranial magnetic stimulation (LF-rTMS) to ameliorate post-stroke abnormal gait and applied wearable sensor technology to analyze patients’ gait, aiming to ascertain the clinical efficacy. Previous preliminary studies [15,16] have suggested that LF-rTMS may enhance walking abilities and motor function in post-stroke patients, making gait patterns more symmetrical and yielding positive effects on balance and postural control. Extensive research has demonstrated that the asymmetry in the cerebral hemispheres following a stroke further impairs the affected hemisphere. A reduction in this asymmetry correlates with improved gait recovery [17], consistent with the central regulation theory of stroke rehabilitation, specifically the interhemispheric competition model. LF-rTMS can restore balance between the hemispheres by inhibiting excitability in the unaffected hemisphere’s corticospinal tract while simultaneously enhancing excitability in the affected hemisphere, thereby improving lower-limb function post stroke [16,18]. LF-rTMS can increase gamma-aminobutyric acid (GABA) release, reduce glutamate release to modulate neurotransmitter levels, promote dendritic plasticity and axonal regeneration, and enhance neural plasticity [19,20,21], facilitating the functional rebuilding and regeneration of damaged neural networks to improve post-stroke lower-limb function. In this study, the LF-rTMS treatment group exhibited significant improvements across various metrics after the treatment, compared to the pseudo-stimulation and control groups. Specifically, improvements were noted in areas such as the FMA-LE, BBS, and MBI, along with detailed gait parameters including the gait cycle, stance phase time, swing phase time, stride length, step height, circle radius, dorsiflexion angle, and walking speed. The gait cycle improved from a pre-treatment average of 2.05 ± 0.51 s to 1.02 ± 0.11 s post treatment (*p* < 0.001). Similarly, significant enhancements were observed in the dorsiflexion angle, increasing from 6.65 ± 1.21 degrees to 18.47 ± 1.06 degrees (*p* < 0.001), and walking speed, which improved from 35.95 ± 7.14 cm/s to 75.03 ± 11.36 cm/s (*p* < 0.001). These quantitative outcomes are comprehensively detailed in Table 3, Table 5 and Table 6, highlighting the clinical efficacy of LF-rTMS in enhancing gait dynamics and overall motor function in post-stroke rehabilitation. The efficacy of LF-rTMS was substantiated through further analysis using Tukey’s HSD post hoc test.

In current clinical settings, gait analysis typically relies on subjective and qualitative methods, such as therapist observation and patient self-reporting [22,23]. While severe gait abnormalities may be perceptible to the naked eye, subtle variations could be overlooked without quantitative measurements [24]. Furthermore, these methods often involve significant inter- and intra-observer variability, thereby impacting disease staging, severity assessment, and subsequent treatment planning. Therefore, this study aims to comprehensively analyze the clinical efficacy of wearable sensor technology in assessing walking impairments in post-stroke patients following TMS. The study involved placing IMU sensors on the lateral aspect of the ankle and utilizing gait-cycle-segmented data to generate time-domain features for classification [12]. Patients were tasked with wearing IMUs and walking back and forth over a 10-m distance, enabling the recording of gait data for a comprehensive biomechanical evaluation. This thorough measurement encompassed the gait cycle, stance phase time, swing phase time, stride length, step height, circumference of movement, dorsiflexion angle, and walking speed, providing a more comprehensive understanding of patients’ walking biomechanics to assess improvements in walking function and offer specific guidance for rehabilitation interventions. The results revealed significant improvements in the gait parameters of the LF-rTMS group, sham stimulation group, and control group following treatment, with the LF-rTMS group showing more pronounced improvement. This indicates that TMS therapy can facilitate the normalization of patients’ gait, enhancing walking stability and coordination. In intergroup comparisons, the LF-rTMS group exhibited a significantly greater improvement in the FMA-LE, BBS, MBI scores and gait parameters compared to the sham stimulation group and control group. This improvement is possibly related to the regulatory effect of LF-rTMS on the M1 area, either by increasing cortical excitability in the affected hemisphere or inhibiting excitability in the unaffected hemisphere to promote functional recovery. Additionally, the sham stimulation group exhibited some improvements in its gait parameters compared to the control group; for instance, the gait speed increased from 34.62 ± 8.71 cm/s to 58.85 ± 9.87 cm/s, and the dorsiflexion angle increased from 6.45 ± 0.77 degrees to 13.65 ± 1.01 degrees. However, these differences were not statistically significant, indicating a potential but not definitive impact of TMS on gait.

While LF-rTMS has previously been demonstrated to improve lower-limb function post stroke, the innovation of our study lies in the application of sensor-based gait evaluation systems, which offer a more objective method of assessing rehabilitation outcomes [17,25]. In comparison, a study [26] utilized three-dimensional gait analysis and also documented significant improvements in the spatiotemporal parameters and joint motion angles of patients with post-stroke walking dysfunction. Specifically, this study noted increases in the stride length, stride frequency, and swing phase percentage on the affected side, alongside reductions in the gait cycle and stance-phase percentage on the involved side. The LF-rTMS group in that study displayed enhanced efficacy, closely aligning with our findings, which similarly showed improvements in the stride length (from 0.56 ± 0.04 m to 0.97 ± 0.08 m), gait speed (from 35.95 ± 7.14 cm/s to 75.03 ± 11.36 cm/s), and a reduction in gait cycle time (from 2.05 ± 0.51 s to 1.02 ± 0.11 s). These results underline the potential of LF-rTMS to significantly enhance the rehabilitation outcomes for post-stroke patients when paired with precise, sensor-based gait analysis tools. Analysis using Tukey’s HSD post hoc test, as reported in Table 7 and Table 8, confirms that these differences between the groups are significant. By integrating such technologies into routine clinical practice, rehabilitation protocols could be tailored more effectively to individual patient needs, potentially accelerating recovery times and improving patients’ quality of life.

This study presents several limitations that merit consideration when interpreting the findings. Firstly, the relatively low sample size may limit the generalizability of the results. While the findings are indicative, a larger cohort would provide a more robust validation of the conclusions and potentially uncover subtle effects not observable with smaller sample sizes. Secondly, the issue of spontaneous recovery in stroke patients, which typically occurs most significantly within the first six months post stroke, was considered despite the presence of a control group. This control group was intended to account for natural recovery processes, allowing for the distinction between the effects of the intervention and natural progression. However, the overlapping of natural recovery and treatment effects can complicate the attribution of improvements, potentially biasing the perceived effectiveness of the intervention. Acknowledging this overlap is crucial for a realistic interpretation of the treatment’s impact. Furthermore, this study focused on the short-term efficacy of the intervention without addressing the long-term sustainability of the benefits. The durability of treatment effects is a critical aspect of stroke rehabilitation, as improvements observed immediately post treatment may not necessarily translate into long-lasting recovery benefits. Factors such as the plateauing of improvements, the risk of rehospitalization, and the potential for secondary conditions can adversely affect the sustained improvement of motor functions and balance. Moreover, the maintenance of gains typically requires ongoing rehabilitation, which may not be feasible for all patients due to various constraints. Future research should thus not only consider larger and more diverse populations to enhance generalizability, but also extend the follow-up period to examine the long-term efficacy of treatments. Additionally, studies exploring methods to support sustained improvements, such as community-based programs or adaptive technologies, would be valuable in addressing the challenges of long-term rehabilitation.

## 5. Conclusions

This study analyzed the clinical efficacy of a wearable sensor-based gait analysis system following transcranial magnetic stimulation treatment for walking impairments in post-stroke patients. The results demonstrated that after six weeks of treatment, the LF-rTMS group exhibited significant improvements in its FMA-LE, BBS, MBI scores, gait cycle, stance phase time, swing phase time, stride length, step height, circumference of movement, dorsiflexion angle, and walking speed compared to the pre-treatment and post-treatment sham stimulation and control groups. This suggests that LF-rTMS can effectively enhance the gait, balance, and quality of daily life of post-stroke patients, improving their walking ability without any observed adverse events during treatment. Research on the impact of LF-rTMS on walking function following stroke is limited; however, this study suggests that LF-rTMS on the unaffected side holds promise as a rehabilitative treatment for improving gait in stroke patients. The application of a wearable sensor-based gait analysis system in this study facilitated the collection and analysis of gait parameters in stroke patients before and after treatment, providing a convenient, refined, accurate, and objective alternative to traditional gait assessments, promising excellent clinical application prospects.

This study has several limitations. Firstly, the sample size was small and limited to the subacute phase of stroke patients; thus, the results cannot be generalized to stroke patients in the acute or chronic phases. The spontaneous recovery and the underlying complexity of stroke heterogeneity are more pronounced in acute and subacute ischemic stroke patients, necessitating further research involving more patients. Secondly, the ideal stimulation parameters and target points for rTMS represent a critical challenge in its application, as these parameters have a significant impact on clinical efficacy. In fact, rTMS as a novel non-invasive neuroregulation technology is still under continual research in its application to clinical conditions, and its principles and mechanisms related to lower-limb functional rehabilitation remain unclear. There is also a lack of consensus regarding the selection of stimulation intensity, duration, and location, necessitating further research.

## Figures and Tables

**Figure 1 sensors-24-03051-f001:**
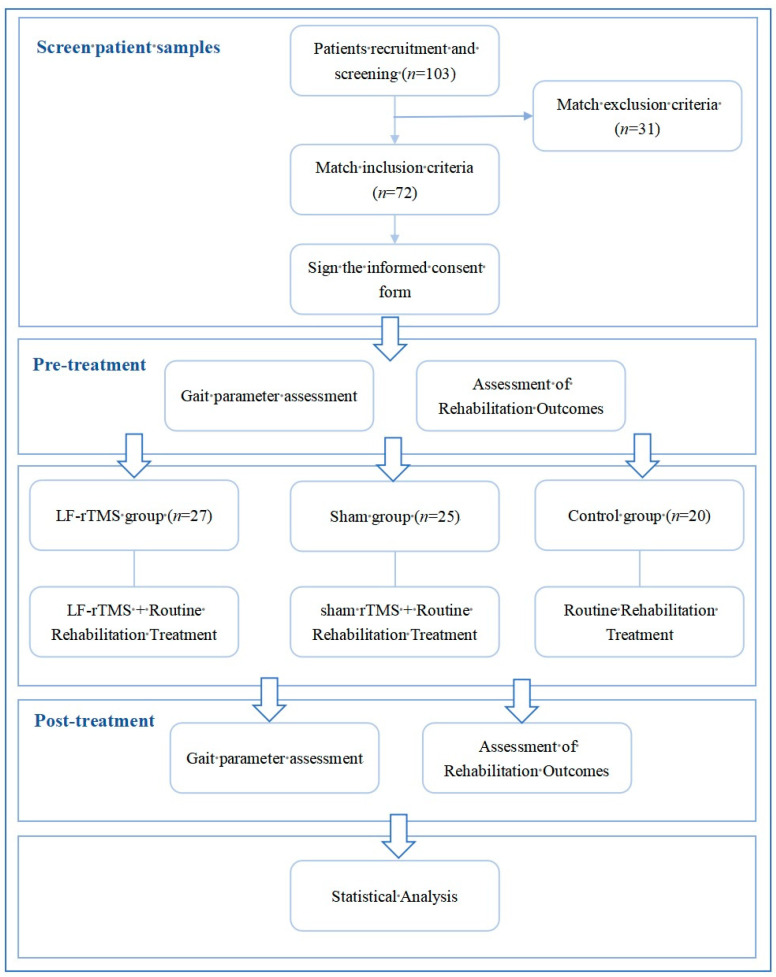
Clinical trial flow chart.

**Figure 2 sensors-24-03051-f002:**
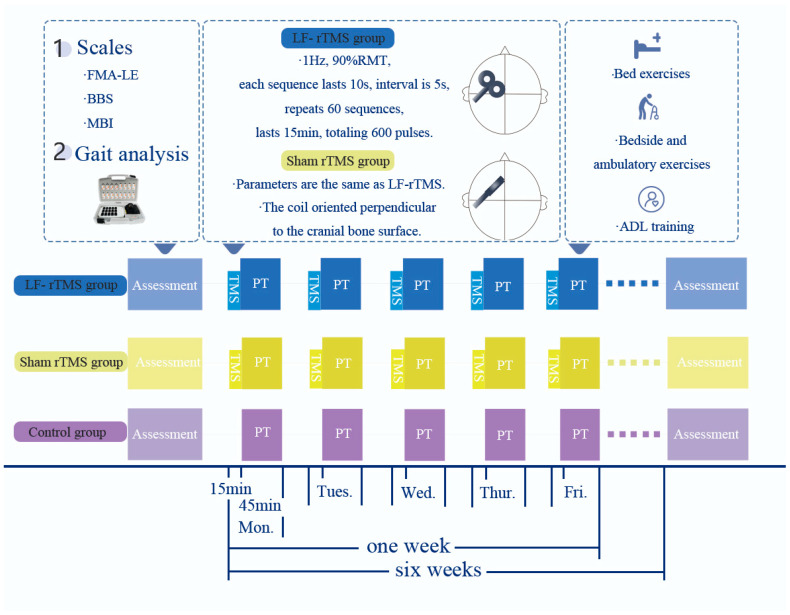
Experimental design drawing. FMA-LE = The Fugl-Meyer Assessment for Lower Extremity; BBS = The Berg Balance Scale; MBI = the Modified Barthel Index; PT = physical therapy, which refers to basic rehabilitation training; ADL = Activity of Daily Living.

**Figure 3 sensors-24-03051-f003:**
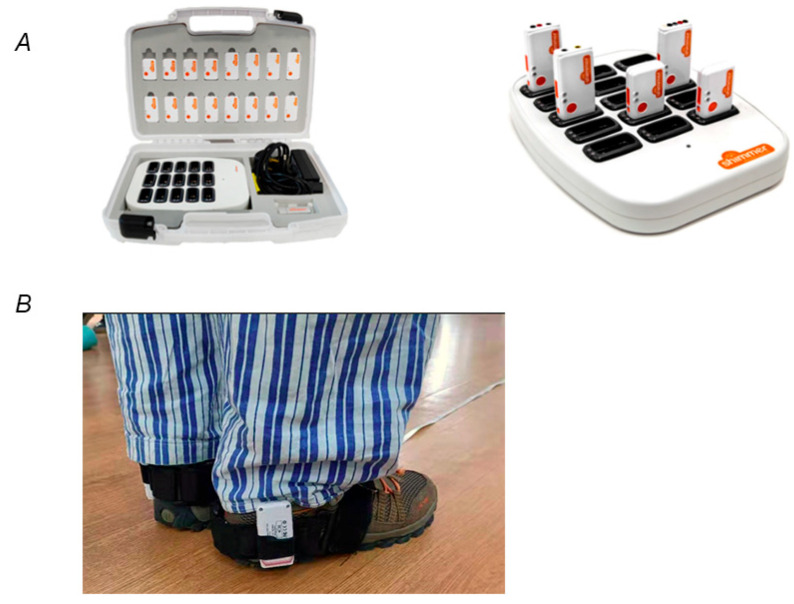
Gait analysis device. (**A**) Consensys Bundle Development kit; (**B**) The IMU sensor-wearing part.

**Figure 4 sensors-24-03051-f004:**
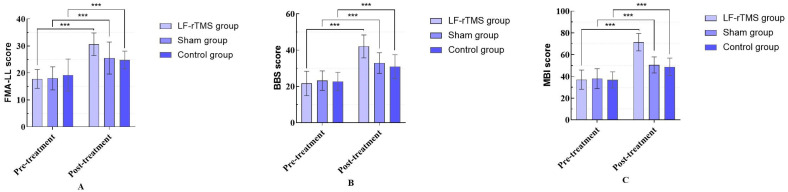
FMA-LE, BBS and MBI were compared among the three groups before and after treatment. (“***” means *p* < 0.001). (**A**) FMA-LE; (**B**) BBS; (**C**) MBI.

**Figure 5 sensors-24-03051-f005:**
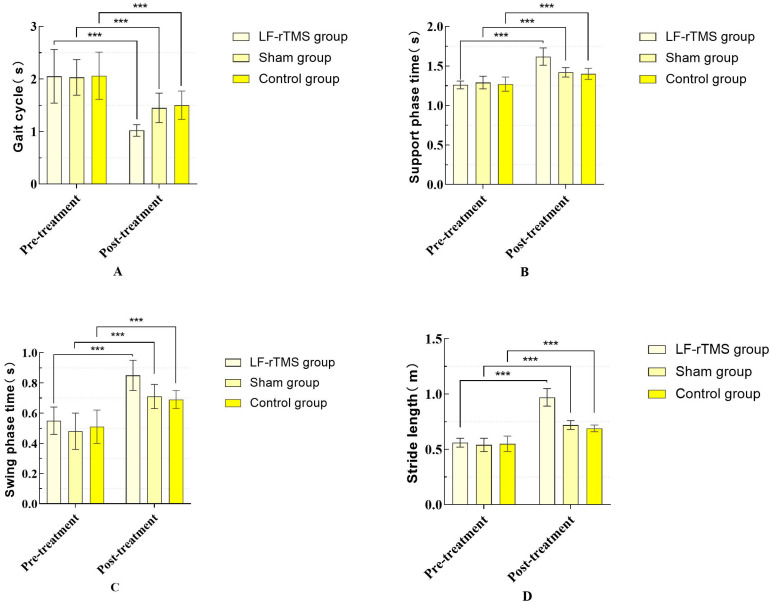
Comparison of the gait cycle, support phase time, swing phase time, and stride length before and after treatment among the three groups of patients. (“***” means *p* < 0.001). (**A**) Gait cycle; (**B**) Support phase time; (**C**) Swing phase time; (**D**) Stride length.

**Figure 6 sensors-24-03051-f006:**
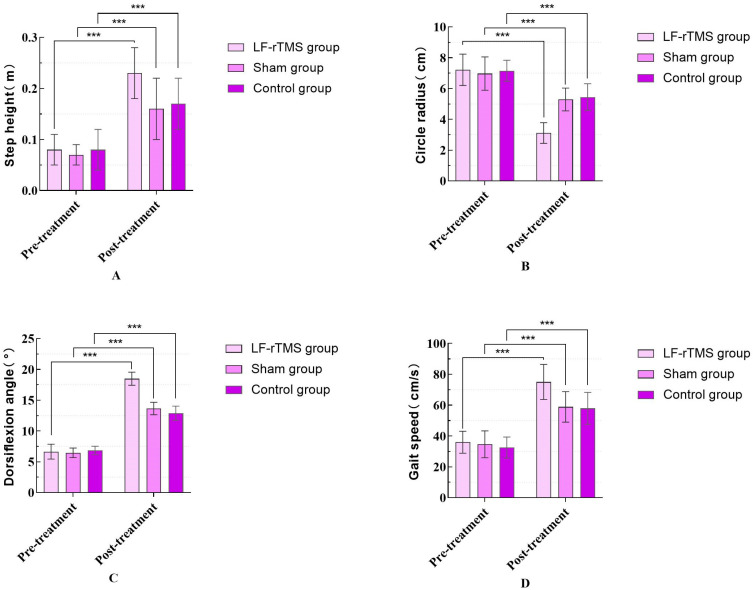
Comparison of the step height, circle radius, dorsiflexion angle, and gait speed before and after treatment among the three groups of patients. (“***” means *p* < 0.001). (**A**) Step height; (**B**) Circle radius; (**C**) Dorsiflexion angle; (**D**) Gait speed.

**Table 1 sensors-24-03051-t001:** Comparison of general information among the three groups of patients.

Group	LF-rTMS Group	Sham Group	Control Group
No.	27	25	20
Gender/No. (%)			
male	11 (41)	8 (32)	6 (30)
female	16 (59)	17 (68)	14 (70)
Age, mean (SD), year	60.95 ± 4.11	61.76 ± 5.04	60.34 ± 6.07
course of disease, mean (SD), day	12.37 ± 6.15	16.42 ± 4.71	15.86 ± 6.59
Type/No. (%)			
Cerebral infarcion	19 (70)	16 (64)	14 (70)
Cerebral hemorrhage	8 (30)	9 (36)	6 (30)
Hemiplegic side/No. (%)			
right	15 (56)	12 (48)	9 (45)
left	12 (44)	13 (52)	11 (55)

**Table 2 sensors-24-03051-t002:** Specifications of Shimmer 3 sensor.

	Accelerometer	Gyroscope	Magnetometer
Range	+16 g	+2000 dps	+49.152 gauss
Sensitivity	1000 LSB/g	131 LSB/dps	667 LSB/gauss
Sampling frequency	400 Hz	400 Hz	400 Hz

**Table 3 sensors-24-03051-t003:** Comparison of FMA-LE, BBS, and MBI among the three groups of patients before and after treatment ($\bar{x}$ ± s).

Group	FMA-LL Score		BBS Score		MBI Score	
	Pre-Treatment	Post-Treatment	Pre-Treatment	Post-Treatment	Pre-Treatment	Post-Treatment
LF-rTMS group	17.78 ± 3.52	30.64 ± 4.17	21.64 ± 6.71	42.04 ± 6.31	36.97 ± 8.79	71.44 ± 8.01
Sham group	18.01 ± 4.28	25.53 ± 5.95	23.18 ± 5.34	32.87 ± 5.67	37.89 ± 9.18	50.52 ± 7.38
Control group	19.23 ± 5.94	24.82 ± 3.27	22.74 ± 4.97	30.91 ± 6.56	36.76 ± 7.42	48.79 ± 7.97
F	0.396	11.952	0.578	17.641	0.366	13.434
*p*	0.547	<0.001	0.381	<0.001	0.587	<0.001

**Table 4 sensors-24-03051-t004:** Results of Tukey’s HSD post hoc test between groups for FMA-LE, BBS, and MBI. A = LF-rTMS group, B = Sham group, C = Control group.

	FMA-LL		BBS		MBI	
	Mean Difference	*p*	Mean Difference	*p*	Mean Difference	*p*
A–B	1.2667	0.5244	16.2667	<0.0001	21.37	<0.0001
A–C	7.4000	<0.0001	17.1667	<0.0001	26.43	<0.0001
B–C	6.1333	<0.0001	0.9	0.799	5.07	0.0002

**Table 5 sensors-24-03051-t005:** Comparison of the gait cycle, support phase time, swing phase time, and stride length before and after treatment among the three groups of patients ($\bar{x}$ ± s).

Group	Gait Cycle (s)		Support Phase Time (s)		Swing Phase Time (s)		Stride Length (m)	
	Pre-Treatment	Post-Treatment	Pre-Treatment	Post-Treatment	Pre-Treatment	Post-Treatment	Pre-Treatment	Post-Treatment
LF-rTMS group	2.05 ± 0.51	1.02 ± 0.11	1.26 ± 0.05	1.62 ± 0.11	0.55 ± 0.09	0.85 ± 0.10	0.56 ± 0.04	0.97 ± 0.08
Sham group	2.03 ± 0.34	1.45 ± 0.28	1.29 ± 0.08	1.42 ± 0.06	0.48 ± 0.12	0.71 ± 0.08	0.54 ± 0.06	0.72 ± 0.04
Control group	2.06 ± 0.45	1.50 ± 0.27	1.27 ± 0.09	1.40 ± 0.07	0.51 ± 0.11	0.69 ± 0.06	0.55 ± 0.07	0.69 ± 0.03
F	0.168	12.689	0.587	12.571	0.354	13.746	0.612	12.075
*p*	0.574	<0.001	0.478	<0.001	0.679	<0.001	0.347	<0.001

**Table 6 sensors-24-03051-t006:** Comparison of the step height, circle radius, dorsiflexion angle, and gait speed before and after treatment among the three groups of patients ($\bar{x}$ ± s).

Group	Step Height (m)		Circle Radius (cm)		Dorsiflexion Angle (°)		Gait Speed (cm/s)	
	Pre-Treatment	Post-Treatment	Pre-Treatment	Post-Treatment	Pre-Treatment	Post-Treatment	Pre-Treatment	Post-Treatment
LF-rTMS group	0.08 ± 0.03	0.23 ± 0.05	7.21 ± 1.02	3.11 ± 0.67	6.65 ± 1.21	18.47 ± 1.06	35.95 ± 7.14	75.03 ± 11.36
Sham group	0.07 ± 0.02	0.16 ± 0.06	6.97 ± 1.08	5.29 ± 0.74	6.45 ± 0.77	13.65 ± 1.01	34.62 ± 8.71	58.85 ± 9.87
Control group	0.08 ± 0.04	0.17 ± 0.05	7.15 ± 0.68	5.44 ± 0.87	6.84 ± 0.67	12.87 ± 1.15	32.47 ± 6.82	57.91 ± 10.35
F	0.245	9.341	0.624	10.715	0.216	9.618	0.971	12.040
*p*	0.657	<0.001	0.478	<0.001	0.579	<0.001	0.488	<0.001

**Table 7 sensors-24-03051-t007:** Results of Tukey’s HSD post hoc test between groups for the gait cycle, support phase time, swing phase time and stride length. A = LF-rTMS group, B = Sham group, C = Control group.

	Gait Cycle (s)		Support Phase Time (s)		Swing Phase Time (s)		Stride Length (m)	
	Mean Difference	*p*	Mean Difference	*p*	Mean Difference	*p*	Mean Difference	*p*
A–B	0.3107	<0.0001	0.0863	0.0326	0.170	<0.0001	0.3037	<0.0001
A–C	0.358	<0.0001	0.2483	<0.0001	0.218	<0.0001	0.3207	<0.0001
B–C	0.0473	0.3704	0.162	<0.0001	0.048	0.0996	0.017	0.7569

**Table 8 sensors-24-03051-t008:** Results of Tukey’s HSD post hoc test between groups for the step height, circle radius, dorsiflexion angle and gait speed. A = LF-rTMS group, B = Sham group, C = Control group.

	Step Height (m)		Circle Radius (cm)		Dorsiflexion Angle (°)		Gait Speed (cm/s)	
	Mean Difference	*p*	Mean Difference	*p*	Mean Difference	*p*	Mean Difference	*p*
A–B	0.0457	0.0004	−2.0147	<0.0001	5.018	<0.0001	18.468	<0.0001
A–C	0.056	<0.0001	−2.0247	<0.0001	6.095	<0.0001	18.864	<0.0001
B–C	0.0103	0.6346	−0.01	0.9916	1.077	0.0001	0.397	0.8108

## Data Availability

If you would like the datasets from this study, please contact the corresponding author.

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
