# Peer review of "Clinical Effect Analysis of Wearable Sensor Technology-Based Gait Function Analysis in Post-Transcranial Magnetic Stimulation Stroke Patients"

_sensors, 2024, doi:10.3390/s24103051_

Round 1

Reviewer 1 Report

Comments and Suggestions for Authors

Review Report for the Manuscript with title: Clinical Effect Analysis of Wearable Sensor Technology-Based Gait Function Analysis in Post-Transcranial Magnetic Stimulation Stroke Patients by authors: Litong Wang, Likai Wang, Zhan Wang, Fei Gao, Jingyi Wu, Hong Tang

The manuscript presents a study that investigates the effectiveness of low-frequency repetitive transcranial magnetic stimulation (LF-rTMS) combined with wearable sensor technology to improve gait functions in post-stroke hemiplegic patients. This work is highly relevant to the fields of neurorehabilitation and medical technology. The topic is timely, and the approach is innovative, embedding advanced therapeutic techniques with cutting-edge technology.

The methodology is well-structured with a robust design involving randomization and control groups. However, the description of methods could be enhanced by:

Providing more detailed parameters of LF-rTMS, such as frequency, intensity, and coil placement.

Expanding on the specifications and placement protocols of the wearable sensors used for gait analysis.

The results are clearly beneficial and show significant findings. Yet, the presentation can be improved by Including more detailed statistical data, such as exact p-values, confidence intervals, and effect sizes.

Enhancing visual representation through additional graphs and tables to better depict the changes observed.

The discussion effectively ties the findings to the broader context of stroke rehabilitation but could be enriched by comparing these results more thoroughly with those of other studies to highlight similarities or differences and discussing the implications of these findings for clinical practice and how they might influence existing treatment protocols.

While some limitations are acknowledged, a more detailed discussion on the impact of these limitations on the study's generalizability and potential biases would be beneficial.

Major Revisions: Enhancements in the methodology and results presentation are needed before the study can be published.

Minor Revisions:  Additional discussion on the clinical implications and limitations are recommended.

Overall, the manuscript offers significant contributions to the field of stroke rehabilitation. The innovative use of LF-rTMS with wearable sensors presents a promising area for future research. With the suggested revisions, this paper can potentially impact clinical practices and offer substantial benefits to stroke survivors.

Kind regards

Reviewer 2 Report

Comments and Suggestions for Authors

This is an interesting study to determine the effects of LF-rTMS on improving gait in post-stroke patients. Among other methods, inertial sensors were used for the objective evaluation of gait quality-related parameters. The results are very interesting, as is the topic. However, there are some points to improve in the document, especially in the methodology, which is not clear. The following are these observations.

· In the abstract: Add quantitative data to the results. Which of the scales showed the best results?

· Line 27: It is desirable to add statistics on the number of patients. What is a high incidence? What are the mortality rates?

· The sentence in lines 47-48 is not correct. Correct it. A gait analysis system does not refer to an AI-assisted system. There are different technologies for gait analysis and, therefore, different systems, such as optical, magnetic, inertial, etc., and all of them are known as gait analysis systems. The reference also does not seem to be the best to support what is mentioned.

· In Table 1, add the data for the control group.

· The registration of the test protocol ChiCTR2300069403 is related to another article titled: "Transcutaneous auricular vagus nerve stimulation on upper limb motor function with stroke: a functional near-infrared spectroscopy pilot study". How are these studies related? Were they the same patients? If so, how could vagus nerve stimulation affect the results?

· Lines 115 to 121: Provide a rationale for the selection of parameters for this study (frequency, duration, power, position, coil type, etc.).

· Line 134: It is not clear; three types of Shimmer sensors are mentioned: IMU, ECG, and EMG. However, it is not mentioned where and for what purpose the EMG and ECG sensors were used. According to the text, it is understood that they only used IMUs, one on each ankle. The description of this instrumentation should be improved.

· The methodology is not clear. When were the stimulations performed, when were the exercises done, and when were the measurements taken? Add a timeline to clarify the times and repetitions of each activity. How long did each stimulation intervention last? How long were the exercises? Much information is missing in this regard.

· In lines 212 to 215, it is suggested to add numerical values indicating the improvements in the mentioned areas, as it is not possible to determine these values with certainty in the figures.

· In lines 224 to 226, data that should be in the methodology are mentioned, indicating how the data were collected.

· Line 251: Mention the improvement numerically. Avoid superlatives.

Reviewer 3 Report

Comments and Suggestions for Authors

The introduction should be expended a bit, please cite prevalence and incidence of stroke since Sensors is not a clinical journal the readers do not have idea of these numbers. 

How did the authors computed the sample size (if any), how did they end up including 72 patients?

The results should be better presented! In Table 1 please also indicate the control group and test for potential groups differences.

The results should be presented in a Table (pre-post differences) not only in the graphics.

Limitations should be in the dicsussion, not in conclusion. While I fully agree concernign the (relatively) low sample size I did not understand why the authors discuss of spontaneuous recovery while they have a control group?

The authors only analysis short term efficacy, they should at least mention and discuss potnetial challenges related to long term efficacy.

Comments on the Quality of English Language

Fine

Round 2

Reviewer 1 Report

Comments and Suggestions for Authors

Dear authors, 

The revised manuscript has substantially addressed the initial concerns raised during the first review. You have made commendable efforts to enhance the manuscript's clarity, depth, and scientific rigor. The enhancements made have significantly improved the manuscript, making it a valuable contribution to the field of stroke rehabilitation. The study is well-positioned to aid clinicians in understanding and implementing wearable sensor technology in conjunction with rTMS for enhancing post-stroke recovery. I propose the acceptance of the article in the present form. 

Best regards

Author Response

We deeply appreciate the time, effort, and expertise that you have contributed to the review of our manuscript. These insightful comments and suggestions have been invaluable in enhancing the quality and clarity of our paper.

Reviewer 3 Report

Comments and Suggestions for Authors

Dear autors,

Thank you for providing a new version of this manuscript. The results are now beter presented. However, looking at the different figures and tables we see that there are statistically significant improvement for the three groups (effect of conventional rehabilitation and spontaneous recovery). However what the authors want to highlight is the added value of rTMS. Therefore, instead of using pre-post test comparison in the different group, they should modelise the difference (post-pre) and see if these differences are statistically different for the three group (ANOVA), alternatively authors could use mix-effect model to integrate both time and group in one single analysis.

Comments on the Quality of English Language

Fine

Author Response

Response to Reviewer:

Thank you once again for your valuable suggestions on improving the statistical analysis in our manuscript. We have taken your feedback into consideration and made significant changes to our statistical methods to more accurately assess the differences and interactions within our data.

We have updated the analysis in the following way:

Pairwise Comparisons: To address the comparisons more robustly, we have revised our approach. Pre hoc pairwise comparisons were now examined using the LSD test to ensure an appropriate balance between Type I and Type II errors in multiple comparisons involving baseline data. For post hoc pairwise comparisons, which are critical for evaluating treatment effects, we have employed Tukey's HSD test. This test is particularly suited for handling multiple group comparisons by controlling the family-wise error rate more stringently.
Statistical Significance: We continue to consider a p-value < 0.05 as statistically significant. This threshold is standard and ensures that our findings are both statistically robust and relevant.
These changes are detailed in the revised methods section of our manuscript and are reflected in the updated results. We believe that these modifications not only adhere to statistical best practices but also enhance the clarity and reliability of our findings, highlighting the specific contribution of rTMS in our study.

We appreciate your guidance, which has been crucial in refining our approach and ensuring that our study's results are presented with the highest level of scientific rigor.